# A Continuous Single-Layer Discrete Tiling System for Online Detection of Corn Impurities and Breakage Rates

**Kun Wu, Min Zhang \*** 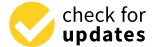**, Gang Wang, Xu Chen and Jun Wu**

Nanjing Institute of Agricultural Mechanization, Ministry of Agriculture and Rural Affairs,
Nanjing 210014, China; 82101195258@caas.cn (K.W.); wanggang32@caas.cn (G.W.); chenxu35@caas.cn (X.C.);
wujun@caas.cn (J.W.)
\* Correspondence: zhangmin01@caas.cn

**Abstract:** In order to improve the accuracy and efficiency of the methods that are used for the detection of impurities in and the breakage rate of harvested corn grains, we propose a classification and identification method using a feature threshold and a backpropagation (BP) neural network that is based on a genetic algorithm. We also constructed a continuous single-layer discrete tile detection system for application to harvested corn grains containing impurities and broken kernels. We conducted an evaluation of the proposed approach with a three-factor and three-level orthogonal experimental design. By setting the working parameters, we realized the continuous single-layer discrete tiling of the grains and 50 grain materials were collected on average in a single picture. In the static test, the error between the system monitoring value and the manual detection value was small, the maximum absolute errors of the breakage and impurity rates were 2.16% and 1.03%, and the average time that was required for each image recognition was 1.71 s. In the experimental environment, the maximum absolute error values of the breakage and impurity rates were 3.48% and 1.78%. The system's identification accuracy and processing time meet the requirements of the online detection of corn characteristics in grain harvesting.

**Keywords:** corn; impurity content; broken rate; online detection; system; test

## 1. Introduction

Online methods that are used to detect the impurity and breakage rate of harvested corn grains provide a basis for realizing the real-time adjustment of system parameters such as threshing and cleaning in the process of combined corn harvesting [1]. Ears of corn often exhibit a high moisture content during the harvesting period and the working parameters of harvesters may be inappropriate, depending on the growing conditions. Existing methods that are used to collect quality information about a harvest are limited in that key components cannot be adjusted in real time according to the information that has been received, resulting in an increased breakage rate during threshing and the inclusion of impurities when the corn is directly harvested. The problem of high breakage rates is notably challenging. In recent years, dramatic breakthroughs have advanced the state of the art in grain inclusion detection and identification technology [2], especially in terms of image feature recognition. Methods to rapidly identify grain crushing and impurities have been developed along with a variety of machine vision-based grain inclusion detection methods. An online detection system for miscellaneous crushing can provide information support for the online real-time automatic adjustment of the working parameters of harvester systems.

Considerable effort has been invested into research on technology that can be used to perform the online detection of impurities in grain-crushing. Chen [3] et al. designed a grain flow collection device that can be used to reduce the adhesion and stacking of the grains during detection and set the appropriate RGB color and area feature thresholds in order to

realize the online identification of impurities and broken grains. Their test results showed that the comprehensive evaluation indices of broken grains, rice stalk impurities, and rice stalk impurities reached 92.92%, 90.65% and 90.52%, respectively. Shaozhong [4] et al. used an optimized watershed segmentation method in order to segment the adhering grains and they performed online identification of the unhulled, whole, and broken buckwheat, using a backpropagation (BP) neural network, which took 4.79 s to process and identify an image. Yang [5] et al. studied the discrete sampling of corn kernels by controlling the gap between two baffles and verified the discrete effect of the sampling device through a simulated experimental platform test. Chen [6] et al. used the intermittent sampling of rice grains and used a decision tree algorithm in order to identify grain impurities and they obtained an accuracy of 76% in their impurity identification. Momin [7] et al. used an HIS color model to segment an image, obtained the area and roundness feature combinations of the objects that were to be examined through image preprocessing, and thus detected the quality of soybean harvests. Their results showed identification accuracies for split beans, contaminated beans, and defective beans and stems/pods of 96%, 75%, and 98%, respectively. However, the existing intermittent or simple slippery discrete sampling methods do not completely solve the problems of material stacking and occlusion due to random and uncontrollable sampling amounts, which affects the detection accuracy and increases the difficulty that is encountered in the use of image recognition algorithms. Therefore, the development of methods that may be used to solve the material bonding, accumulation, and occlusion that occur during material sampling and optimize image recognition algorithms may be considered as effective measures to improve the efficiency and accuracy of crushed grain and impurity identification.

In this study, we propose a continuous single-layer discrete tiling sampling device. By optimizing the parameter combination of the feeding speed of the sampling mechanism, the speed of the conveyor belt, and the opening gap, the blocking and bonding of the sample material can be synchronized in order to realize single-layer discrete tiling. We also propose an image recognition model that can be used to analyze the obvious differences between the image features of corn kernels and impurities, which first separates the kernels and impurities and then judges the integrity of the kernels. Moreover, we have constructed a regression model that is designed to predict the quality of grain materials, established a calculation method for the impurity content and breakage rate, and formulated a calculation and processing system for the impurity content and breakage rate. The proposed approach is designed to perform rapid and accurate identification of the impurity content and breakage rate of a combine harvest of corn and display the information that has been obtained. We experimentally evaluated the recognition accuracy and processing time of the system and the results verified its effectiveness. This research can provide information to support the subsequent automatic adjustment of the working parameters of threshing and cleaning systems of corn combine harvesters.

## 2. Materials and Methods

### 2.1. The Overall Structure and Working Principle of the System

The presently described discrete tiling-type online detection system for corn harvest grain impurity rate and breakage rate and the system's structure are shown in Figure 1. As shown, the system consists of two parts, including the hardware and software, of which the discrete tiling sampling device is shown in Figure 2, including an external sheaf mechanism, conveying mechanism, outer frame of the sampling device, LED light group, image acquisition mechanism, industrial computer, power system, and variable-voltage power supply. The fixed discharge mechanism is composed of a model 57H2P5442A4-PG stepping motor, a DM542 driver, a CS10-3 controller, and a 6-slot outer sheaf with a diameter of 100 mm. The maximum output torque of the motor is 10 N·m, the conveyance mechanism is composed of a DC motor, a gear rod with a length of 120 mm, a pure black rubber synchronous belt, and the maximum output torque of the DC motor is 5 N·m. The

software component comprises the image processing, grain and impurity identification, and the calculation model of impurity content and breakage rate based on MATLAB software.

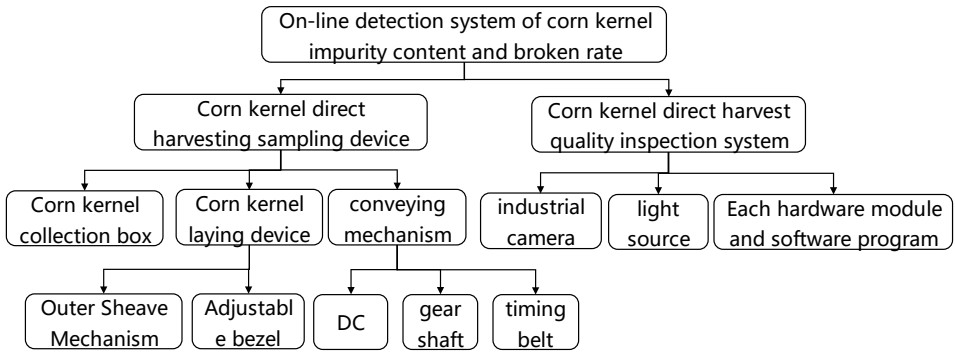

**Figure 1.** The structure of the monitoring system for harvested corn grains with impurities and breakage rates.

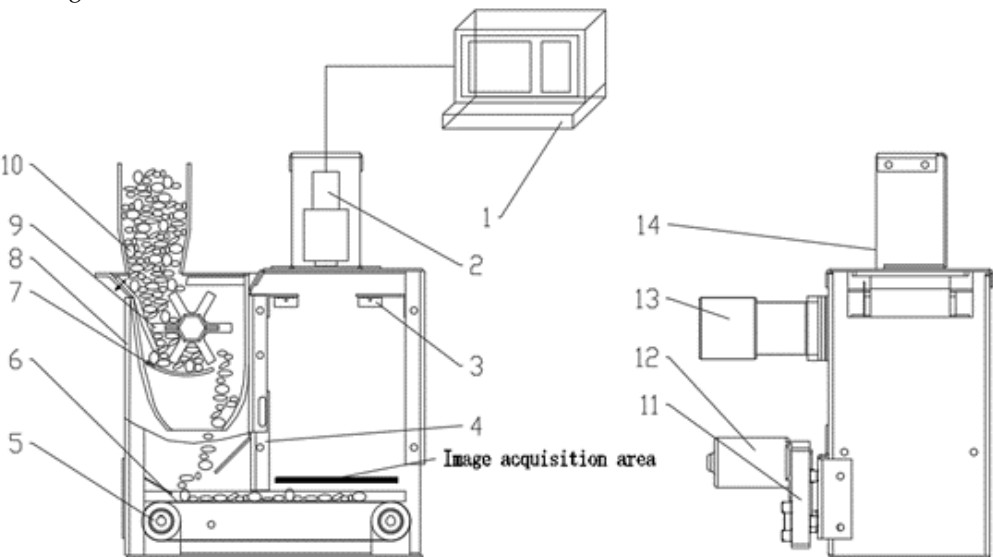

**Figure 2.** Discrete tile sampling mechanism for corn harvested kernels. (1) Industrial computer, (2) industrial camera, (3) LED light source, (4) adjustable baffle, (5) gear shaft, (6) timing belt, (7) discharge tongue, (8) collection box, (9) outer sheaf, (10) corn kernels, (11) reducer, (12) DC motor, (13) stepper motor, and (14) camera fixing shell.

The image acquisition system mainly includes an industrial computer, a Daheng Mercury series (MERCURY) camera, and a dual LED light source. The specific model of the camera is Daheng's MER-231-41U3MC, the lens that was used is an optional 12 mm zoom lens, the camera's working distance is 180 mm, and the accuracy was found to be less than 0.1 mm. It was equipped with a two-channel light source controller that was used to control the double-bar LED light groups. The spatial dimensions of the entire corn harvesting grain sampling device are 360 mm × 290 mm × 270 mm.

Figure 3 shows a flowchart of the operation of the system. Materials (i.e., complete corn kernels, broken kernels, impurities, etc.) enter the discrete tile sampling device from the feeding inlet above the sheaf and the stepper motor drives the sheaf to rotate in order to realize the controllable feeding of the harvested corn kernels. The sheaf slides under the blocking of the sliding plates on both sides such that the material falls into the conveyor belt evenly and gently and this prevents splashing and accumulation under the synchronous belt. The motor drives the conveyor belt and the adjustable baffle structure controls the single-layer passage of the material in order to prevent stacking. The material passes through the continuous-tiling sampling mechanism in order to realize the required single-layer discrete tiling and it then enters the image acquisition area. The camera is then

used to collect the sample image under the double-LED supplementary light. The image information is transmitted to the industrial computer for image processing in order to realize the classification and identification of the complete grains, impurities, and broken grains that are present in the image. Then, the impurity content and breakage rates of the collected images are calculated and the results are displayed on the output interface in real time.

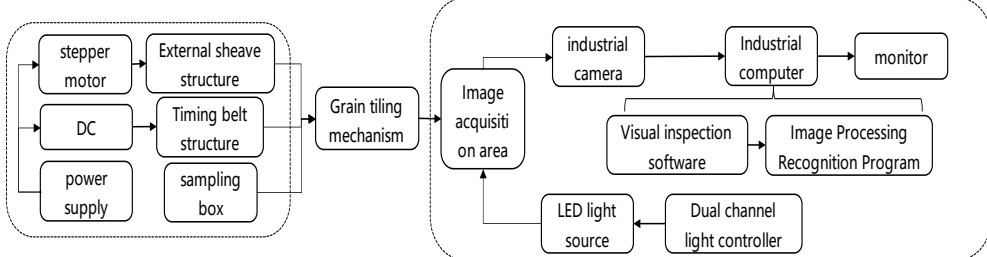

**Figure 3.** Workflow of the monitoring system.

According to the national standard GB/T5494-2019, the mass of a small inspection sample of harvested corn grains containing impurities and breakage is about 100 g. The designed image collection area was 120 mm × 120 mm and a single image captured about 16 g of grains. Therefore, it was necessary to continuously collect 5–6 sample images and carry out cumulative calculation.

### 2.2. Optimization of Working Parameters of Discrete Tiled Sampling Mechanism

The outer sheaf mechanism and the conveying mechanism of the sampling device are driven by two motors. The matching of the outer sheaf speed and the speed of the conveyor belt is key to realizing the effect of material tiling. If the outer sheaf speed was too fast, an excessive amount of the material would accumulate. If the conveyor belt speed was too fast, the acquired image would be distorted and the amount of material in a single image would be small. The need to increase the number of images that are collected led to the increase of the system's processing time [8].

In order to obtain a reasonable speed-matching relationship, a three-factor and three-level orthogonal test optimization analysis was carried out on the speed of the sheaf A, the speed of the conveyor belt B, and the adjustment baffle clearance C [9]. Taking the material image recognition accuracy rate $P_v$ and the material conveying flow rate as the test evaluation index Q, a total of 9 groups of tests were carried out and the test levels of the experimental factors are shown in Table 1.

**Table 1.** Factor level table of orthogonal tests.

| Level | Sheaf Speed A (r/min) | Conveyor Belt Speed B (r/min) | Adjust the Baffle Clearance C/mm |
|:---:|:---:|:---:|:---:|
| 1 | 10 | 69.2 | 10 |
| 2 | 7.5 | 58.1 | 8 |
| 3 | 5 | 46 | 6 |

There were obvious differences in the geometric size of the grains at different positions on the corn cobs. The thickness of the 30 corn grains that were obtained from different sections varied from 3 to 6 mm and a reasonable gap was set.

The identification accuracy rate of the materials refers to the ratio of the number of complete grains, broken grains, and impurities that were identified in the collected images to the total number of grains in the image; the conveying flow rate represents the mass of the conveyed materials per unit of time. The expressions of these two values are given below.

$$P_v = \frac{T}{T_o} \times 100\%,\tag{1}$$

$$Q = \frac{M}{t} \times 100\%. \tag{2}$$

In the formulas, T is the accurate number of target objects that were identified, $T_o$ is the total number of materials that were collected in the picture, $P_v$ is the recognition accuracy rate, the mass of materials that were collected within t seconds is M grams, and Q is the material flow that was conveyed.

In the orthogonal experiment design scheme, A, B, and C refer to the speed of the sheaf wheel, the speed of the conveyor belt, and the gap from adjusting the baffle, which are the three main factors. In addition, Z1 and Z2 represent the accuracy of the grain identification and grain conveying flow, respectively. The experimental protocol and results are shown in Table 2.

**Table 2.** Test scheme and results.

| Serial Number | Sheaf Speed A | Conveyor Belt Speed B | Adjust the Baffle Clearance | Grain Recognition Accuracy Z1 | Grain Conveying Flow Z2/(g/s) |
|---|---|---|---|---|---|
| 1 | 1 | 2 | 2 | 88.69% | 9.90 |
| 2 | 1 | 1 | 1 | 92.93% | 9.37 |
| 3 | 2 | 1 | 3 | 92.52% | 8.20 |
| 4 | 2 | 3 | 2 | 93.09% | 7.57 |
| 5 | 3 | 3 | 1 | 92.62% | 6.27 |
| 6 | 3 | 2 | 3 | 93.43% | 5.41 |
| 7 | 3 | 1 | 2 | 92.00% | 3.67 |
| 8 | 1 | 3 | 3 | 85.47% | 9.22 |
| 9 | 2 | 2 | 1 | 90.33% | 6.38 |

A range analysis was carried out from the orthogonal test results of the combination of the 3 test factors and 3 levels in Table 2 and the range analysis data that are shown in Table 3 were obtained. The order of the factors affecting the recognition accuracy and material conveying flow, from primary to tertiary, was A > B > C. That is, the influence of the speed of the sheaf on the two indicators was the largest, followed by the influence of the speed of the conveyor belt. The least influential factor was the adjustment of the baffle's clearance. The detection process requires high recognition accuracy and a large grain conveying flow. The analysis showed that the optimal combination of rotational speed matching with the highest recognition accuracy was A3B1 and the optimal rotational speed matching that would obtain the highest grain conveying flow rate was A1B3. The ultimate goal of the experiment was to obtain a higher recognition accuracy rate while comprehensively considering various indicators and to select the speed of the sheaf to be controlled at 5 r/min and the speed of the synchronous belt to be controlled at 69.2 r/min in order to achieve a higher level of recognition accuracy.

**Table 3.** Range analysis.

| Index | | Sheaf Speed A | Conveyor Belt Speed B | Adjust Baffle Clearance |
|---|---|---|---|---|
| Recognition accuracy Z1 | 1 | 89.03% | 92.48% | 91.96% |
| | 2 | 91.98% | 90.82% | 91.26% |
| | 3 | 92.68% | 90.39% | 90.47% |
| | Range R | 3.65% | 2.09% | 1.49% |
| | Factor, primary and secondary | | | A, B, C |
| Grain conveying flow Z2 (g/s) | 1 | 9.49 | 7.08 | 7.34 |
| | 2 | 7.38 | 7.23 | 7.05 |
| | 3 | 5.12 | 7.69 | 7.61 |
| | Range R | 4.37 | 0.61 | 0.56 |
| | Factor, primary and secondary | | | A, B, C |

By adjusting the baffle's gap in order to affect the accuracy of the grain identification process and the one-way ANOVA of the grain-conveying flow, the *p* values were all greater than the significance level of 0.05 and it was concluded that the effects of different gaps on the two results did not differ significantly. In order to prevent the stacking and clogging of grains that can be caused by factors such as vibration, the gap of the adjustment baffle was set to 10 mm.

### 2.3. Design of Image Acquisition and Processing System

### 2.3.1. Optimization of Light Source Brightness Parameters

Light sources are an important part of machine vision systems [10,11] and they are a necessary condition for such systems to collect images of a high level of quality. When an industrial camera acquires a picture of a sampled material, the brightness of the fill light affects the quality of the image acquisition and further affects the recognition accuracy [12,13]. In the proposed approach, a test analyzes the quality of the images that are collected with the use of LED lamps under different brightness settings and the best fill light conditions are then able to be selected.

Then, the working parameters of the industrial camera are set and the brightness of the LED lights is controlled and adjusted. Five sets of grain images were collected and grayscale image processing was performed on them. A graph of this process that was used for the analysis is shown below. Figure 4 shows the information of harvested corn grains under different brightness conditions.

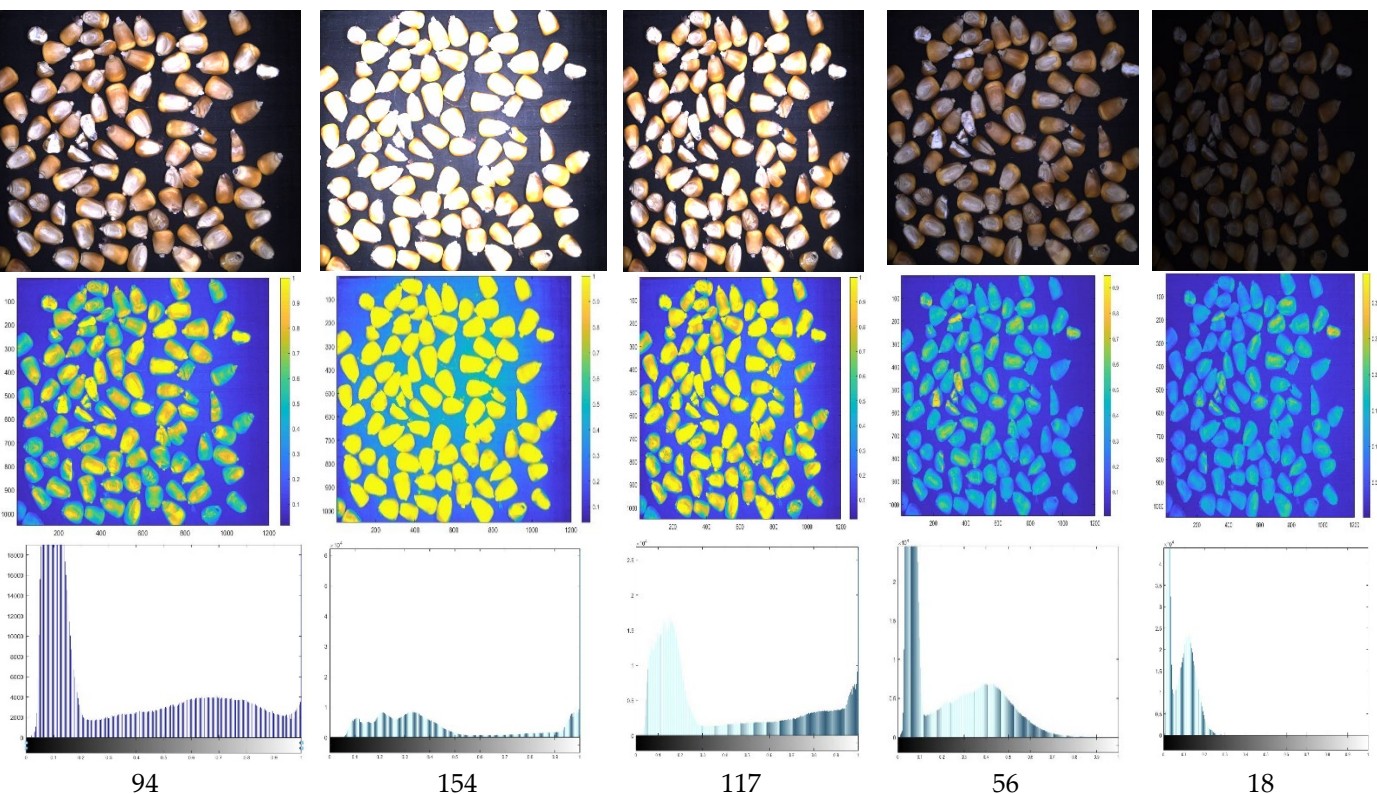

**Figure 4.** Grain image and grayscale histogram under different brightness conditions.

By thresholding the background and the target object, the gray values of the double-type transformation were 0.4, 0.35, 0.3, and 0.25 as the brightness threshold limit and the pixel distribution probability of the brightness value was counted in order to obtain the data that are provided in Table 4.

**Table 4.** Distribution probabilities of luminance values for different threshold boundaries of images with different luminances.

| Threshold Interval | Grayscale Mean 94/% | Grayscale Mean 154/% | Gray Mean 117/% | Grayscale Mean 56/% | Gray Mean 18/% |
|---|---|---|---|---|---|
| [0, 0.4] | 58.93 | 43.31 | 54.45 | 77.80 | 100 |
| [0, 0.35] | 56.55 | 36.47 | 53.02 | 71.31 | 100 |
| [0, 0.3] | 54.33 | 28.26 | 51.68 | 65.61 | 99.99 |
| [0, 0.25] | 52.46 | 28.89 | 49.70 | 60.52 | 99.84 |

From the data that are outlined in the provided figures and tables, it may be observed that when the average grayscale value of the collected sample images was 94 the probability of being in the corresponding threshold interval (that is, the background grayscale threshold interval) was not considerably different and both were greater than 50%. From the grayscale histogram, the corn kernel information and the background can be separated well and the color was found to be saturated and true. When the fill light's intensity was too high or too weak, separating the background and the sampled material information became difficult, the brightness threshold was weak, and the color was distorted resulting in the excessive segmentation of the edge of the collected image and the distortion of the image information, which eventually affected the identification accuracy for impurities and broken kernels.

### 2.3.2. Image Preprocessing

An industrial camera was used to acquire an image of a single layer of the material, as shown in Figure 5a, and the image preprocessing was performed [14]. After the image's grayscale processing, an arithmetical mean filter was used to perform a smoothing and opening operation in order to remove the noise information in the picture, as shown in Figure 5b. The Otsu threshold segmentation process was performed in order to obtain a binary image, as shown in Figure 5c. The optimized watershed segmentation algorithm was used to segment the stuck corn kernels. By reconstructing the gray image and obtaining the image's gray gradient as the segmentation function, the minimum correction gradient magnitude image was imposed between the background and the object and the initial segmentation of the grains' adhesion was realized based on the watershed segmentation algorithm. The R + G color of the original image was extracted in order to construct the grayscale gradient, mark the image's background, and select the still-adhered grains for watershed segmentation. The segmented grain image was subjected to boundary suppression and a filling processing in order to prevent the edge grains from being misidentified due to the cropping and processing of the internal details of the grain that generate holes. Then, any small interfering closed areas were removed in order to obtain a complete segmentation image, as shown in Figure 5d.

### 2.3.3. Classification and Identification of Impurities and Grain Integrity

Appearance shape features, color features, and texture features are important indicators that can be used to distinguish corn kernels [15,16]. The preprocessed image was marked with connected regions in order to obtain the feature information of each connected object in the image. In this study, the shape features of the image area, boundary pixels, long axis, short axis, Hu moment invariant 1, and moment invariant 2 were extracted. From these, the R mean, B mean, R variance, H mean, S mean, H variance, I Color features of variance and S variance, texture features of the gray mean, variance, and entropy (a total of 17 feature parameter datapoints) were obtained and the data were studied and analyzed.

The variation ranges of the corn kernels and their impurity characteristics were counted in the proposed approach based on the fact that the first-order invariant moment and the R-B eigenvalue are significantly different and the boundary pixel points and aspect ratio are different. As shown in Table 5, Feature combination thresholding enabled the image pre-classification of the impurities and grains [17].

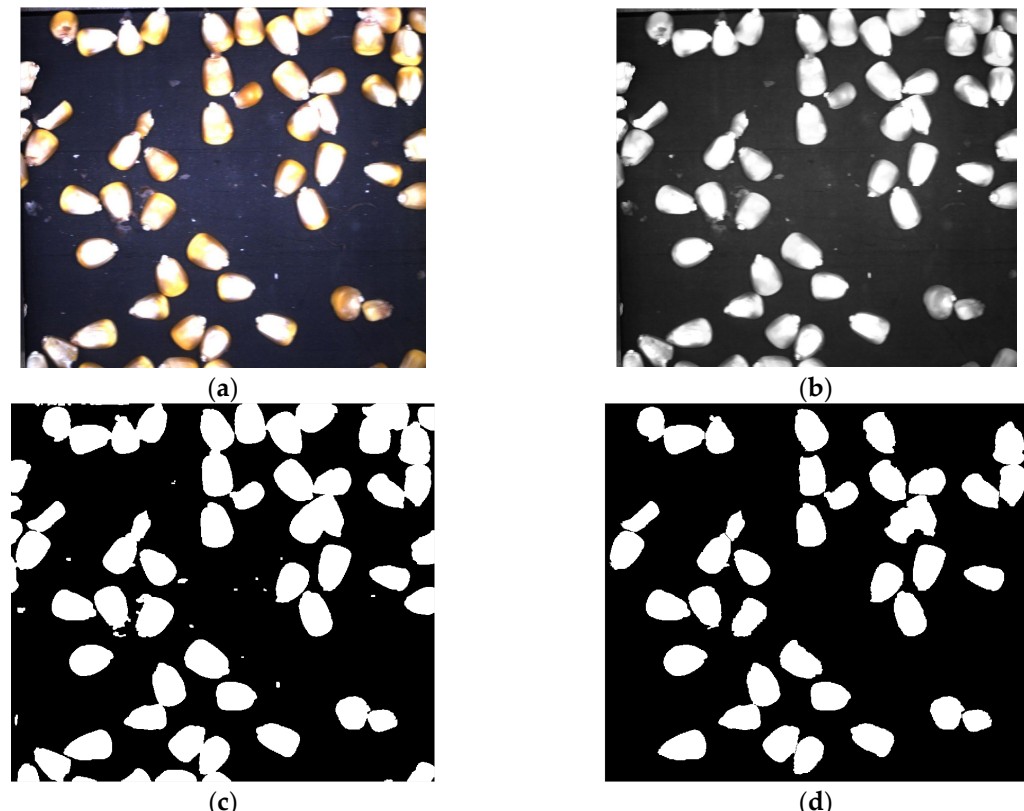

**Figure 5.** Corn kernel harvest image preprocessing. (**a**) Original material collection image. (**b**) Image after filtering and open arithmetical processing. (**c**) Binary image. (**d**) Preprocessed images.

**Table 5.** Statistics of variation range of corn kernel and impurity characteristics and threshold setting.

| The Characteristic Type | Complete Grains | Crush the Grain | Impurity | Threshold Range | Operation Logic |
|---|---|---|---|---|---|
| First-order invariant moment | 0.161~0.215 | 0.160~0.273 | 0.175~0.703 | >0.273 | or |
| R-B value | 0.155~0.496 | 0.007~0.253 | −0.069~0.054 | <0.07 | or |
| Border pixels | 132~329 | 111~404 | 117~458 | >404 | or |
| Length-to-diameter ratio | 1.121~3.023 | 1.046~3.423 | 1.154~8.203 | >3.423 | or |

In order to classify the corn kernels' integrity, the proposed approach uses the GaBP neural network [18] to realize the identification of broken kernels. Using the Levenberg–Marquardt (BpLM algorithm) training model [19], this structure has the advantage of faster computational speed. A 4-layer BP neural network structure was constructed and the weights and thresholds of the BP neural network were iteratively optimized by the use of a genetic algorithm in order to reduce the error of the network training that can be caused by the randomness of the parameters and to avoid falling into local optima. The basic steps of the optimization of the genetic algorithm are shown in Figure 6.

We collected 240 images of complete corn kernels and broken kernels and we then extracted the long axis, short axis, R mean, B mean, R variance, S mean, H variance, S variance, gray mean, variance, entropy, and Hu difference of the grains. A total of 13 features of variable moment 1 and Hu invariant moment 2 were used as the training parameters of the neural network, 360 feature datapoints were used as the training samples

of the neural network, and the remaining 120 datapoints were used as the testing set. The test recognition accuracy rate reached 95%.

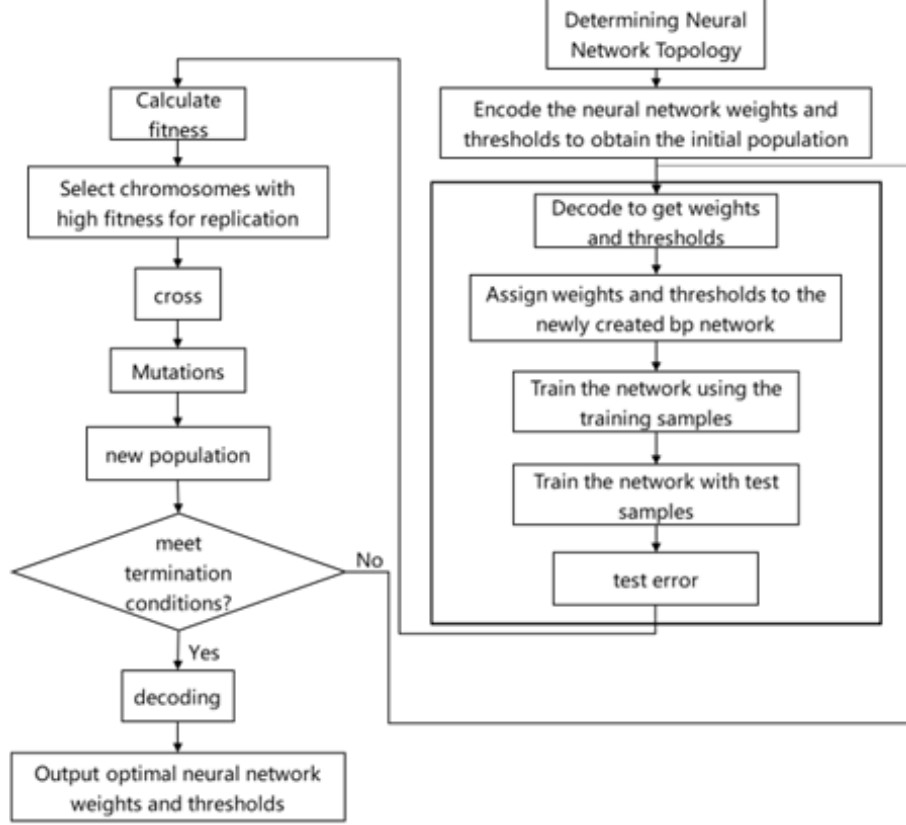

**Figure 6.** Basic steps of genetic algorithm optimization.

*2.4. Calculation of the Impurity Content and Breakage Rate of Harvested Corn Grains*

The indicators of the impurity content and breakage rate of harvested corn grains directly reflect the quality of the directly harvested corn grains [20]. By establishing the pixel area of each substance and its corresponding mass regression model, the pixel area of the intact grain, broken grain, and impurities in the test sample are able to be obtained through automatic identification and the mass of various substances is able to be predicted based on the regression model. Then, the impurity content of the collected sample is able to be calculated with the breakage rate [21,22].

2.4.1. Establishment of Regression Model of Pixel Area and Material Mass

In the experiment, a moisture detector was used to detect the moisture content of the corn grains in the samples and the moisture content of the grain samples was found to be 34.26%. Forty images of complete corn kernels, broken kernels, and impurities were collected continuously and the corresponding mass was recorded. We used MATLAB to perform linear regression fitting on the pixel area and mass value of each recorded image and to draw the linear regression graph, the scatter plot of the original data, and the residual analysis graph accordingly, as shown in Figure 7.

The fitting effect of the various substances was shown to be good through regression analysis and the error of the relationship between the data and the fitting linear relationship was within the allowable range from the residual test. The linear regression coefficients R2 of the complete and broken corn grains and impurities were calculated to be 0.9966, 0.9998, and 0.9975, respectively, all with a strong degree of fitting. The regression

equations corresponding to grain mass and pixel area were established, respectively, as Formulas (3)–(5).

$$M_w = 4.523 \times 10^{-5} \times S_w - 1.081, \tag{3}$$

$$M_p = 3.247 \times 10^{-5} \times S_p - 0.261, \tag{4}$$

$$M_z = 1.373 \times 10^{-5} \times S_z + 0.392. \tag{5}$$

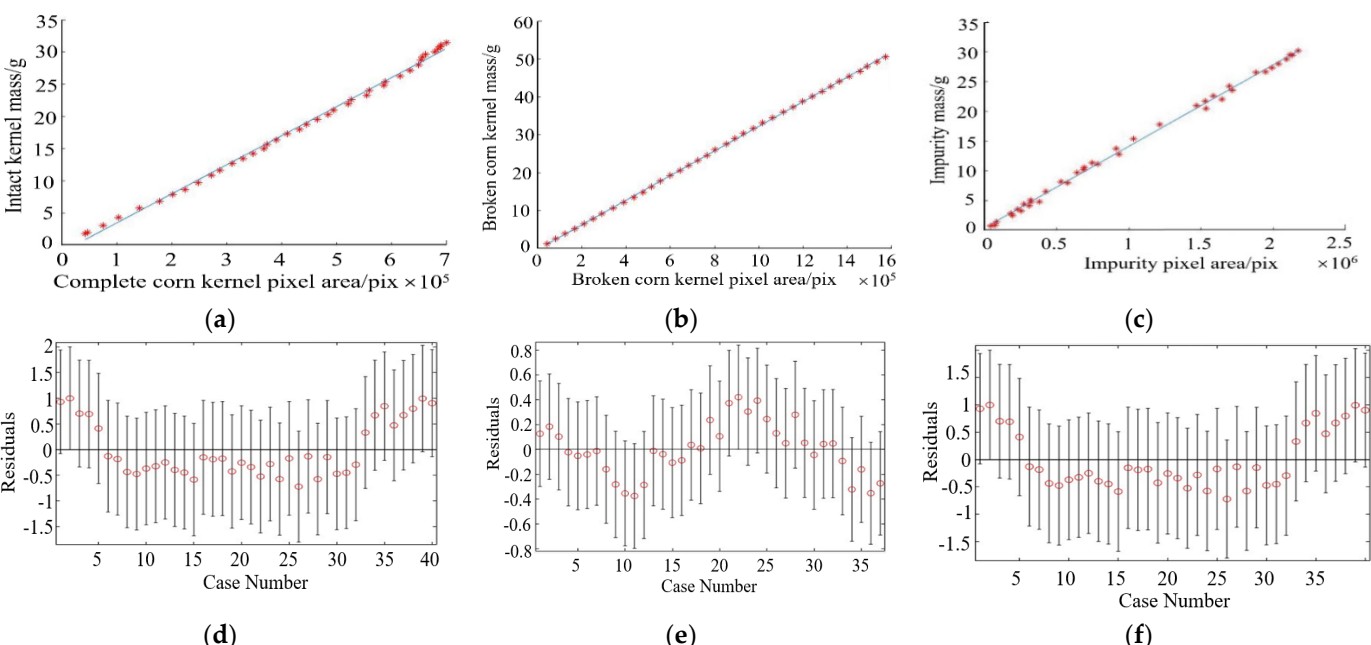

**Figure 7.** Linear regression analysis of pixel area and mass. (**a**) Relationship between the quality of the whole corn kernel and the pixel area, (**b**) relationship between broken corn kernel quality and pixel area (**c**) relationship between impurity quality and pixel area, (**d**) residual analysis of intact corn mass and pixel area, (**e**) residual analysis of broken corn quality and pixel area, and (**f**) residual analysis of impurity mass and pixel area.

In the formulas, $M_w$, $M_p$, $M_z$ are the mass of the intact corn kernels, broken corn kernels, and impurities, respectively, in units of g, and $S_w$, $S_p$, and $S_z$ are the pixel areas corresponding to the intact corn kernels, broken corn kernels, and impurities, respectively, in units of pixels.

When predicting the mass of corn materials of different moisture contents, the linear relationship between the calculated mass and pixel area (6) of a given sample can be used and the regression model can be carried out by using the mass relationship between the corn materials with different moisture contents (7). Coefficient correction is then able to be performed in (8). The formulas are provided as follows.

$$M_i = b_1 \times S_i + b_0, \tag{6}$$

$$\frac{M_b}{M_a} = \frac{(1 - a\%)}{(1 - b\%)}, \tag{7}$$

$$M_b = \frac{(1 - a\%)}{(1 - b\%)} \times (b_1 \times S_a + b_0). \tag{8}$$

In these formulas, the moisture content of sample A is $a\%$, the mass is $M_a$, the moisture content of sample B is $b\%$, its mass is $M_b$, and $b_1$ and $b_0$ are the linear regression coefficients.

By using the mass relationship of the corn material under different moisture contents, and based on the linear regression relationship between the mass of corn material under a certain moisture content and the pixel area, the corresponding correction coefficient

$\frac{(1-a\%)}{(1-b\%)}$ can be obtained in order to finally realize the mass prediction across different moisture contents.

### 2.4.2. Calculation of Impurity Content and Fragmentation Rate

In order to improve the representativeness of the collected samples through the continuous collection of sample images and their identification and classification, 5 images were accumulated in sequence and the mass of the various substances therein was predicted by the use of fitting. Because the regression-fitting equation between the mass and the pixel area tends to deviate from the zero-position interval, when the S area is 0, the mass M becomes negative. In this study, the mass M was corrected to zero in order to avoid the negative value phenomenon. M is the predicted mass of the corresponding substance and the entire calculation process is shown in Figure 8.

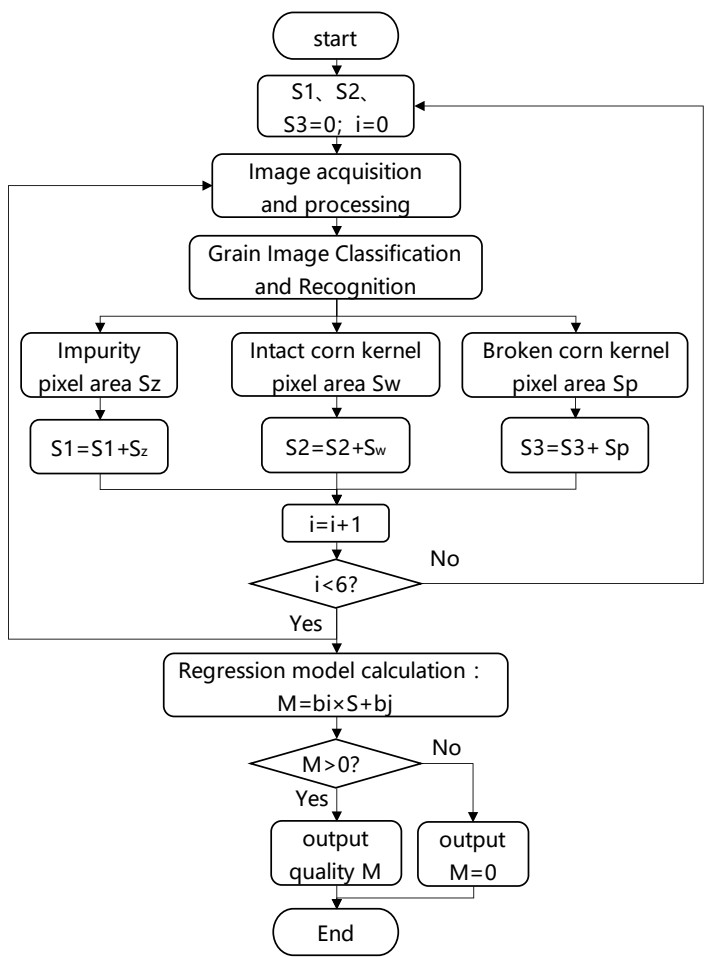

**Figure 8.** Diagram of mass-prediction calculation steps.

The masses of the various substances were obtained through regression fitting prediction and the impurity content and broken rate of the sample harvested corn grains were calculated. The calculation formula is expressed as follows.

$$P_p = \frac{M_p}{M_w + M_p + M_z} \times 100\%, \tag{9}$$

$$P_z = \frac{M_z}{M_w + M_p + M_z} \times 100\%. \tag{10}$$

In the formulas, $P_p$ and $P_z$ are the breakage rate and impurity content and $M_w$, $M_p$, and $M_z$ are the predicted mass of the intact corn kernels, broken corn kernels, and impurities, respectively.

### 3. Test Results and Discussion

*3.1. Classification and Identification of Impurities and Grains*

We adjusted the baffle's clearance in order to obtain the best working condition for the monitoring device. We then continuously collected materials, adjusted the working brightness of the LED, and performed image acquisition and identification. The image acquisition and identification test is shown in Figure 9 and the display interface of the monitoring system is shown in Figure 10. In the proposed system, accurate grain segmentation was achieved by the image preprocessing of the collected material images and the identification of impurities and broken grains was achieved by the application of a feature threshold combination and the GABP neural network. The recognition accuracy was calculated for every three consecutive images in order to ensure the reliability of the evaluation and three sets of recognition results were collected for analysis. Figure 11 shows the image processing and recognition results of the harvested corn grains. The proposed approach first obtains the segmentation map of the harvested corn grain image and it then performs boundary suppression processing, as shown in Figure 11b. Through the step-by-step classification and identification of impurities and broken kernels and their labeling and counting, respectively, the classification and labeling results were plotted, as shown in Figure 11c.

We manually counted the number of impurities, intact grains, and broken grains in order to verify the accuracy of the machine's recognition. The recognition accuracy P of Formula (2) is used to represent the performance of the machine recognition system. It may be observed from the image recognition results in Table 6 that the recognition accuracy rates of the harvested corn grains in the three recognition experiments were 92.5%, 95.35%, and 92.38%, respectively, and that the overall online recognition performance was good.

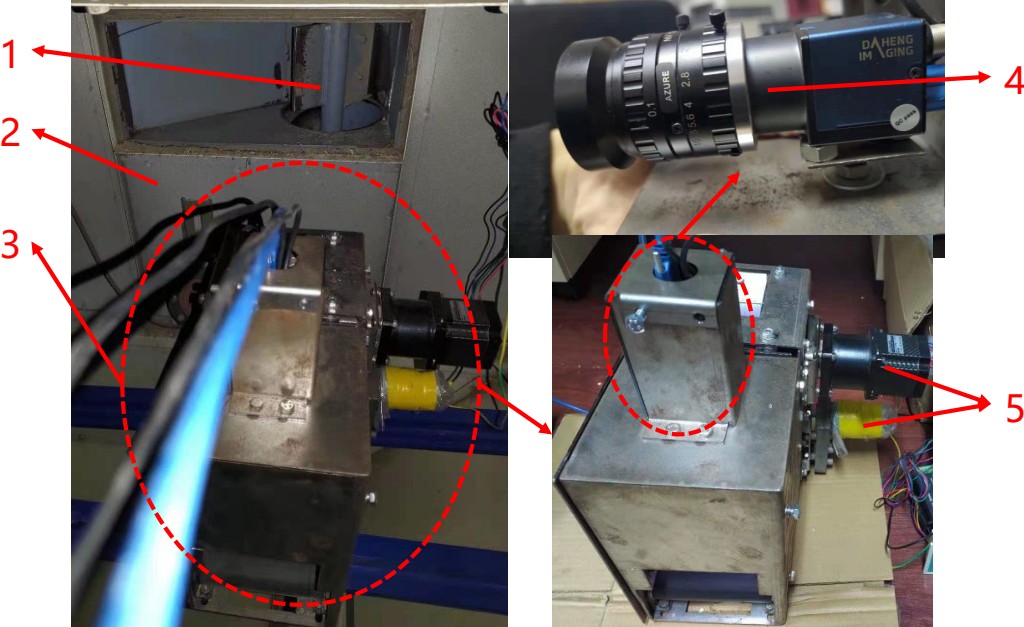

**Figure 9.** Image acquisition and recognition experiment of harvested corn grains. (1) Grain churn, (2) grain tank, (3) corn harvesting grain sampling monitoring device, (4) industrial camera, and (5) motor.

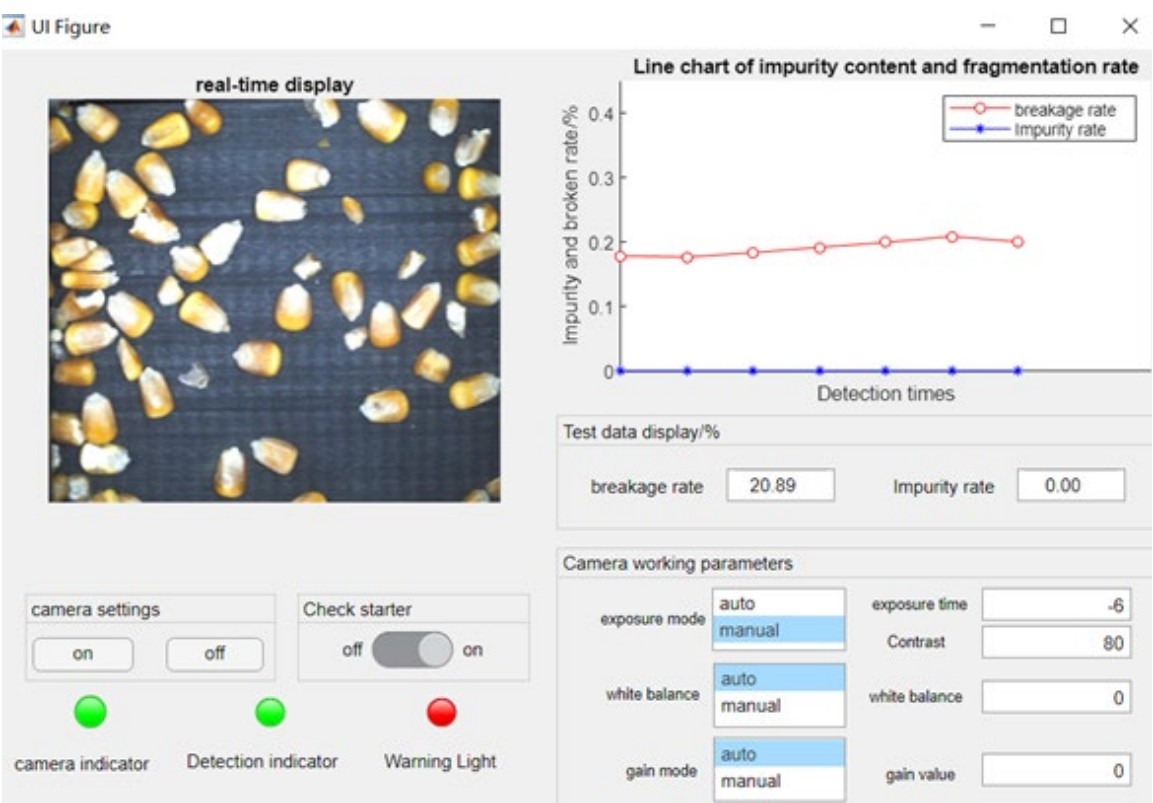

**Figure 10.** Online monitoring software interface.

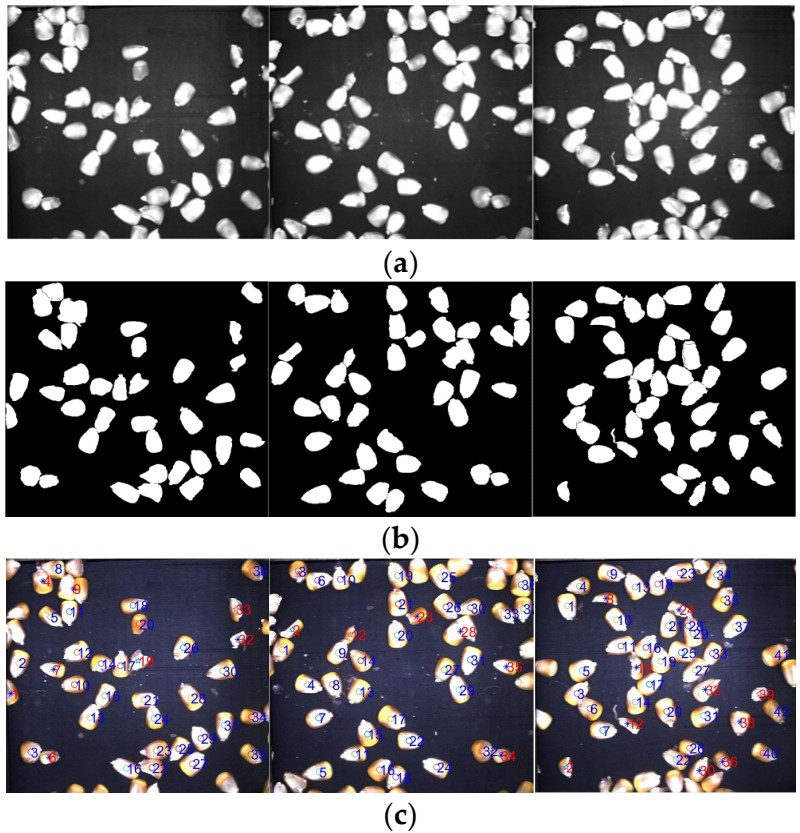

**Figure 11.** Image processing and labeling of harvested corn grains. (**a**) Grayscale image of harvested corn grains. (**b**) Boundary suppression binary map. (**c**) Corn grain harvest image marker.

**Table 6.** Image recognition results of harvested corn kernels.

| Number | Total Number of Grains | Identify the Exact Number | The Number of Misidentifications | Identification Accuracy/% |
|---|---|---|---|---|
| 1 | 120 | 111 | 9 | 92.5 |
| 2 | 129 | 123 | 6 | 95.35 |
| 3 | 105 | 97 | 8 | 92.38 |

*3.2. System Static Test*

In the test, three groups of samples with different impurity contents and fragmentation rates were collected and identified from an image. Each group of samples was continuously collected for five pictures, three consecutive times, and the mass of the various substances was calculated and summed. Statistical calculations of the rate and breakage rate were performed. The total mass, impurity content, and fragmentation rate of the three groups of samples were obtained by the use of manual identification and calculation. The first group of samples had a fragmentation rate of 10.64% and an impurity content of 1.16%; the second group of samples had a fragmentation rate of 10.44% and an impurity content of 1.14%. The third group of samples had a breakage rate of 13.53% and an impurity content of 1.18%. The test data results are shown in Figure 12a,b. All three of the samples were sampled and tested by the online detection system for the impurity rate and breakage rate of the harvested corn grains and a summary of the results is shown in Table 7.

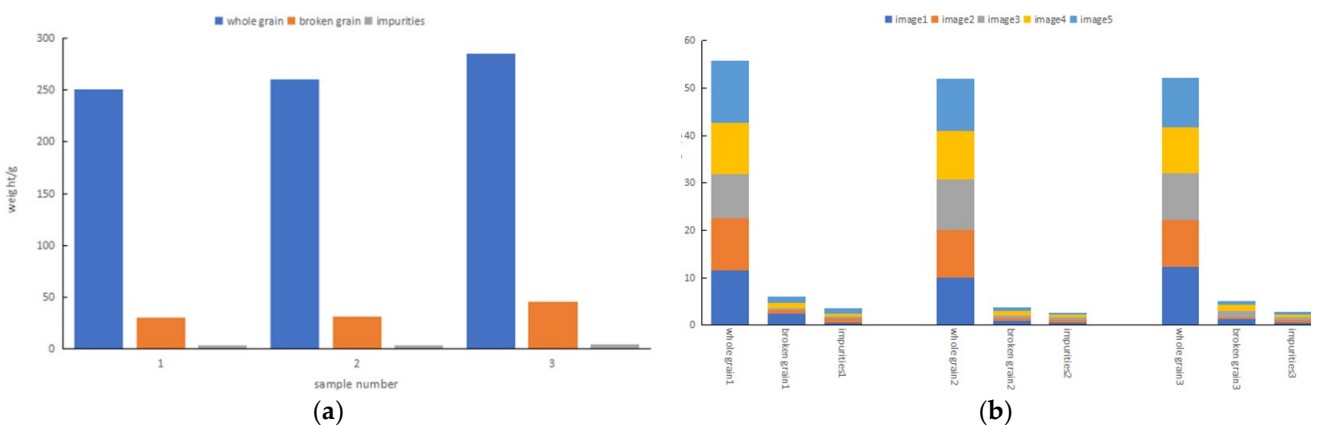

**Figure 12.** Histograms of mass test results of various substances such as corn grains. (**a**) Manual test data results. (**b**) Machine inspection data results.

**Table 7.** Test results of impurity content and breakage rate.

| Group Number | Image Detection | | Manual Inspection | | Crushing Rate Absolute Error/% | Contains Impurity Rate Absolute Error/% | Single Picture Processing Time/s |
|---|---|---|---|---|---|---|---|
| | Crushing Rate/% | Impurity Rate/% | Crushing Rate/% | Impurity Rate/% | | | |
| 1 | 11.48 | 0.79 | 10.64 | 1.16 | 0.84 | 0.37 | 1.67 |
| 2 | 13.09 | 1.74 | 10.44 | 1.14 | 2.65 | 0.6 | 1.76 |
| 3 | 15.69 | 2.21 | 13.53 | 1.18 | 2.16 | 1.03 | 1.71 |
| average value | \ | \ | \ | \ | 1.88 | 0.67 | 1.71 |

The absolute error calculation formula that was used is as follows.

$$\eta = |T_t - T_n|. \tag{11}$$

In the formula, the detected value is the breakage rate or impurity content rate of the image of the harvested corn kernels; whereas, the true value is the breakage rate or

impurity content rate of artificially detected harvested corn kernels and this is the absolute error value.

The comprehensive recognition accuracy of the online recognition method that was applied to broken grains and impurities in rice, designed by Chen and colleagues [3], was over 90% and the recognition speed of a single image therein was 1.86 s. However, their system could not realize the detection of corn harvesting impurities and crushing. The online monitoring system for grain impurity content and breakage rate that was studied by Gu [21] exhibited a recognition accuracy of more than 84.68%. However, because of the push–pull intermittent grain sampling method, the results could only be obtained every 4 s and the recognition efficiency of the entire monitoring system was low. It may be observed from Table 7 that the absolute error between the monitoring values of the impurity rate and the broken rate that was obtained by the online monitoring system that is proposed in this work and the actual manual detection result was small. These results reach the current technical level that is required for the detection of cereal impurity and breakage rates. Moreover, the average processing time of each image was, at most, 1.71 s, which is better than existing state-of-the-art methods. By increasing the collection area and increasing the amount of harvested corn grain images that are collected, the monitoring efficiency of the system can be further improved.

### 3.3. System Real-World Environmental Testing

The sampling device was installed at the entrance of the grain auger box of a corn harvester and multiple tests were carried out on the corn grain materials, which had moisture contents of 12.59% and 13.4%, respectively. We used the monitoring system to calculate the obtained corn harvest's impurity rate and breakage rate and, when the system performed continuous monitoring, we manually collected and detected the materials at the discharge port and subsequently compared and analyzed the monitoring value and the manual detection value. Continuous monitoring was carried out in each experiment; nine groups of data were obtained and three groups were also manually sampled and tested for comparative analysis. The first test was carried out on corn with a moisture content of 12.59% and the data results are shown in Table 8 below; the second test analysis was carried out on corn with a moisture content of 13.4% and the data results are shown in Table 9.

From the data of the two experiments that are detailed in the tables, it may be observed that the average absolute errors of the monitoring of the corn impurity rate and breakage rate of the whole system were 1.84%, 1.09%, 2.12%, and 0.86%. Compared with the static test, the fluctuation of the monitoring data of the whole system was slightly larger, mainly because the whole device was affected by the vibration of the harvester and other interfering factors. In contrast, the impurity content of the sample was small and the distribution was uneven, resulting in a large absolute error of the impurity content. In follow-up research, the system's resistance to the influence of external vibration interference must be further optimized and the system's ability to adapt to the external environment and improve the detection accuracy through measures such as shock absorption or vibration isolation should be further improved.

**Table 8.** The results of the first experiment were dynamically monitored for the yield of harvested maize grains' impurity and crushing rate.

| Number | Sample Group | Image Detection | | Manual Inspection | | Absolute Error of Crushing Rate/% | Contains Impurity Absolute Error/% |
|---|---|---|---|---|---|---|---|
| | | Crushing Rate/% | Impurity Rate/% | Crushing Rate/% | Impurity Rate/% | | |
| 1 | 1 | 7.58 | 1.46 | 8.63 | 0.41 | 1.05 | 1.05 |
| 2 | 1 | 6.82 | 2.02 | 8.63 | 0.41 | 1.81 | 1.61 |
| 3 | 1 | 7.65 | 1.34 | 8.63 | 0.41 | 0.98 | 0.93 |
| 4 | 2 | 6.79 | 1.42 | 8.97 | 0.56 | 2.18 | 0.86 |
| 5 | 2 | 6.58 | 1.63 | 8.97 | 0.56 | 2.39 | 1.07 |

**Table 8.** *Cont.*

| Number | Sample Group | Image Detection | | Manual Inspection | | Absolute Error of Crushing Rate/% | Contains Impurity Absolute Error/% |
| --- | --- | --- | --- | --- | --- | --- | --- |
| | | Crushing Rate/% | Impurity Rate/% | Crushing Rate/% | Impurity Rate/% | | |
| 6 | 2 | 5.95 | 2.34 | 8.97 | 0.56 | 3.02 | 1.78 |
| 7 | 3 | 7.44 | 2.00 | 8.82 | 1.16 | 1.38 | 0.84 |
| 8 | 3 | 5.87 | 2.24 | 8.82 | 1.16 | 2.95 | 1.08 |
| 9 | 3 | 8.06 | 1.71 | 8.82 | 1.16 | 0.76 | 0.55 |
| | average value | \ | \ | \ | \ | 1.84 | 1.09 |

**Table 9.** The results of the second experiment were dynamically monitored by the yield of harvested maize grains' impurity and crushing rate.

| Number | Sample Group | Image Detection | | Manual Inspection | | Absolute Error of Crushing Rate/% | Contains Impurity Absolute Error/% |
| --- | --- | --- | --- | --- | --- | --- | --- |
| | | Crushing Rate/% | Impurity Rate/% | Crushing rate/% | Impurity Rate/% | | |
| 1 | 1 | 19.55 | 1.05 | 22.83 | 1.02 | 3.28 | 0.03 |
| 2 | 1 | 21.12 | 0 | 22.83 | 1.02 | 1.71 | 1.02 |
| 3 | 1 | 20.22 | 2.06 | 22.83 | 1.02 | 2.61 | 1.04 |
| 4 | 2 | 21.15 | 1.29 | 21.92 | 1.13 | 0.77 | 0.16 |
| 5 | 2 | 20.12 | 0 | 21.92 | 1.13 | 1.8 | 1.13 |
| 6 | 2 | 19.7 | 3.13 | 21.92 | 1.13 | 2.22 | 2 |
| 7 | 3 | 22.73 | 0 | 23.24 | 1.34 | 0.51 | 1.34 |
| 8 | 3 | 19.76 | 1.05 | 23.24 | 1.34 | 3.48 | 0.29 |
| 9 | 3 | 20.5 | 0.6 | 23.24 | 1.34 | 2.74 | 0.74 |
| | average value | \ | \ | \ | \ | 2.12 | 0.86 |

## 4. Conclusions

(1) We have proposed a discrete tiling sampling mechanism. We also conducted an experimental optimization and found that the best tiling performance was achieved when the speed of the sheaf and the synchronous belt were 5 r/min and 69.2 r/min, respectively.

(2) By analyzing the influence of LED brightness on image acquisition quality, we concluded that the optimal brightness adjustment range occurred when the average grayscale value of the image was around 94. We have also proposed a step-by-step recognition method for classifying impurities and corn kernels, in which the impurities are removed based on the combination of threshold features and a GABP neural network is used to realize the recognition of the intact and broken corn kernels. The classification and recognition accuracy rates of the proposed approach were over 92.5%.

(3) We have established a regression model comprising material mass and pixel area and the mass prediction model, under testing on materials with different moisture contents, was corrected by the use of the coefficient. In the static test, the error between the impurity rate and the broken rate that were recognized by the system and the manual detection value was small, the maximum absolute error of the breakage rate and the impurity rate was 2.16% and 1.03%, respectively, and the average processing time of each image did not exceed 1.71 s. In the actual dynamic environment, the maximum absolute error values of the breakage rate and impurity rate between the proposed system's identification and manual detection were 3.48% and 1.78%.

The proposed approach uses the continuous collection of grain images to accumulate and calculate the impurity content and breakage rate. The number of grains that are present in a single image affects the cumulative weighting of the data for the images. In future research, we plan to adjust the structure of the device in order to increase the image

collection field of view and increase the number of single images that are collected. We also intend to improve the computational performance of the image processing methods and the calculation efficiency of the system's impurity fragmentation rate.

**Author Contributions:** Conceptualization, M.Z. and K.W.; methodology, M.Z.; software, K.W.; validation, J.W. and G.W.; formal analysis, K.W.; investigation, M.Z.; resources, M.Z.; data curation, K.W.; writing—original draft preparation, K.W.; writing-review and editing, M.Z.; visualization, K.W.; supervision, X.C. project administration, M.Z.; funding acquisition, M.Z. All authors have read and agreed to the published version of the manuscript.

**Funding:** This research was funded by Jiangsu Provincial Agricultural Science and Technology Independent Innovation Fund SCX (20)3031 and institute-level basic scientific research business expenses project of the Chinese Academy of Agricultural Sciences (S202008).

**Institutional Review Board Statement:** Not applicable.

**Informed Consent Statement:** Not applicable.

**Data Availability Statement:** The data presented in this study are available on request from the authors.

**Acknowledgments:** The authors thank the editor and anonymous reviewers for providing helpful suggestions for improving the quality of this manuscript.

**Conflicts of Interest:** The authors declare no conflict of interest.

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
