# Peer review of "A Continuous Single-Layer Discrete Tiling System for Online Detection of Corn Impurities and Breakage Rates"

_agriculture, doi:10.3390/agriculture12070948_

Round 1
Reviewer 1 Report
1. In the right box of Figure 3, there are two sub-boxes connected after the industrial camera. One is the industrial computer and another is the visual inspection software. It makes me confused about which equipment did the visual inspection software and the image processing recognition program. If the two processes were done by the industrial computer, they should not be connected after the industrial camera but after the industrial computer.
2. Why use 94, 154, 117, 56, and 18 as the brightness setting? Also, why is the double-transformed grayscale value is set at 0.4, 0.35, 0.3, and 0.25? How did the author conclude with these setting parameters?
3. On page 9, line 286, where is q in the formula? In line 292, there is no following formula. In lines 293-295, where is the formula? This whole paragraph is very confusing and affects reading.
4. On page 13, line 275, the formula should be in the same format as the others.
5. I suggest that the author hire a professional English editing company to revise the paper.
Author Response
1.In the right box of Figure 3, there are two sub-boxes connected after the industrial camera. One is the industrial computer and another is the visual inspection software. It makes me confused about which equipment did the visual inspection software and the image processing recognition program. If the two processes were done by the industrial computer, they should not be connected after the industrial camera but after the industrial computer.
Response: In this paper, the detection software and the image processing program are all completed by the industrial computer, and finally the monitoring results are output and displayed on the display, and the flow chart has been modified according to the suggestions.
2.Why use 94, 154, 117, 56, and 18 as the brightness setting? Also, why is the double-transformed grayscale value is set at 0.4, 0.35, 0.3, and 0.25? How did the author conclude with these setting parameters?
Response: In the actual environment, the contrast and brightness of the captured image are not only affected by the brightness of a single LED, but also by interference such as natural light. In this paper, the gray value of the image is used to represent the brightness value of the image, which directly reflects the quality of the collected image. By analyzing the grayscale histogram of the image, according to the relationship between the peaks and troughs of the histogram, a threshold is selected and equidistantly reduced to obtain the threshold limit, and then the images are observed and compared.
3.On page 9, line 286, where is q in the formula? In line 292, there is no following formula. In lines 293-295, where is the formula? This whole paragraph is very confusing and affects reading.
Response: For the formula problem, it has been revised as suggested, and the paragraphs of the article have been revised and adjusted.
4.On page 13, line 275, the formula should be in the same format as the others.
Response: The formula format has been modified as suggested.
5.I suggest that the author hire a professional English editing company to revise the paper.
Response: We have sought professional editing. We hope the writing is acceptable for publication now.
Reviewer 2 Report
This is an interesting study, but some important points must be clarified or fixed:
1. There are grammar issues, and long sentences in the abstract Please re-write the abstract again and check the grammar.
2. In the abstract, the research problem must be explained clearly with data. In addition, please summarize the content of the paper; it should not include too much information.
3. In the introduction, show the scientific contribution of this work with concrete data as well as the organization of this paper should be specified at the end of the introduction.
4. In equation 1, How is the value of T (accurate number) determined? How are correctly and incorrectly intact corn kernels identified? How are correctly and incorrectly broken kernels identified? All these four parameters should be used to decide the accuracy, recall, and precision of the grain recognition.
5. The authors should write more details about the BP neural network and genetic algorithms (block diagram, classification results, etc). Why do authors use these techniques?
6. The authors must justify the effectiveness of the proposed method by comparing it with the other baselines.
7. In the conclusion, the authors should specify future works.
Author Response
This is an interesting study, but some important points must be clarified or fixed:
1.There are grammar issues, and long sentences in the abstract Please re-write the abstract again and check the grammar.
Response: We have sought professional editing. We hope the writing is acceptable for publication now.
2.In the abstract, the research problem must be explained clearly with data. In addition, please summarize the content of the paper; it should not include too much information.
Response: The research content has been revised as suggested.
3.In the introduction, show the scientific contribution of this work with concrete data as well as the organization of this paper should be specified at the end of the introduction.
Response: Data has been added to the introduction as requested to illustrate the contribution and significance of the work, and to describe the organization of the article.
4.In equation 1, How is the value of T (accurate number) determined? How are correctly and incorrectly intact corn kernels identified? How are correctly and incorrectly broken kernels identified? All these four parameters should be used to decide the accuracy, recall, and precision of the grain recognition.
Response: In Equation 1, the accuracy of T is determined by identifying and marking the collected images and identifying them manually. In this paper, the sample data of broken and intact kernels are trained based on Bp neural network, and the completeness of the grains collected from the images is classified and identified. This paper mainly analyzes three factors. These three parameters have a great influence on the laying effect of corn kernel sampling. The laying situation directly affects the quality of the collected corn kernel images and then affects the image processing and classification recognition.
5.The authors should write more details about the BP neural network and genetic algorithms (block diagram, classification results, etc). Why do authors use these techniques?
Response: The main research content of this paper is the design of material sampling device, parameter optimization and detection system research. Added Genetic Algorithm details. The Bp neural network has the advantages of simple structure, suitable for dealing with complex classification problems of internal mechanisms, and strong generalization ability. However, the initial weights and thresholds of the Bp neural network are uncertain, which leads to the uncertainty of the training recognition results. By adding the genetic algorithm to optimize the Bp neural network Neural network weights and thresholds make the results of neural network training and testing relatively stable.
6.The authors must justify the effectiveness of the proposed method by comparing it with the other baselines.
Response: There are few researches on the online detection technology of corn kernel impurity broken rate. In this paper, a single-layer discrete tile sampling device is studied and designed, and a step-by-step classification and identification method of kernels and impurities is constructed. In the whole system, the identification efficiency and accuracy of impurities and broken grains have obvious advantages through experiments. The test is conducted in a real environment, and the accuracy is high by comparing with manual test.
7.In the conclusion, the authors should specify future works.
Response: Further research work has been added to the conclusion as suggested.
Round 2
Reviewer 2 Report
Thank you, I don't have any other comments.